# *Yersinia enterocolitica*-Derived Outer Membrane Vesicles Inhibit Initial Stage of Biofilm Formation

**DOI:** 10.3390/microorganisms10122357

**Published:** 2022-11-29

**Authors:** Guoxiang Ma, Yu Ding, Qingping Wu, Jumei Zhang, Ming Liu, Zhi Wang, Zimeng Wang, Shi Wu, Xiaojuan Yang, Ying Li, Xianhu Wei, Juan Wang

**Affiliations:** 1College of Food Science, South China Agricultural University, Guangzhou 510642, China; 2Key Laboratory of Agricultural Microbiomics and Precision Application, Ministry of Agriculture and Rural Affairs, Guangdong Provincial Key Laboratory of Microbial Safety and Health, State Key Laboratory of Applied Microbiology Southern China, Institute of Microbiology, Guangdong Academy of Sciences, Guangzhou 510070, China; 3Department of Food Science and Engineering, Institute of Food Safety and Nutrition, College of Science & Engineering, Jinan University, Guangzhou 510632, China

**Keywords:** outer membrane vesicles, *Yersinia enterocolitica*, biofilm, lipopolysaccharides, temperature

## Abstract

*Yersinia enterocolitica* (*Y. enterocolitica*) is an important food-borne and zoonotic pathogen. It can form biofilm on the surface of food, increasing the risk to food safety. Generally, outer membrane vesicles (OMVs) are spherical nanostructures secreted by Gram-negative bacteria during growth. They play a role in biological processes because they contain biologically active molecules. Several studies have reported that OMVs secreted by various bacteria are associated with the formation of biofilms. However, the interactions between *Y. enterocolitica* OMVs and biofilm are unknown. This study aims to investigate the effect of *Y. enterocolitica* OMVs on biofilm formation. Firstly, OMVs were extracted from *Y. enterocolitica* Y1083, which has a strong biofilm-forming ability, at 15 °C, 28 °C and 37 °C and then characterized. The characterization results showed differences in the yield and protein content of three types of OMVs. Next, by co-culturing the OMVs with *Y. enterocolitica*, it was observed that the OMVs inhibited the initial stage of *Y. enterocolitica* biofilm formation but did not affect the growth of *Y. enterocolitica*. Furthermore, biofilm formation by *Salmonella enteritidis* and *Staphylococcus aureus* were also inhibited by OMVs. Subsequently, it was proved that lipopolysaccharides (LPS) in OMVs inhibited biofilm formation., The proteins, DNA or RNA in OMVs could not inhibit biofilm formation. Bacterial motility and the expression of the biofilm-related genes *pgaABC*, *motB* and *flhBD* were inhibited by LPS. LPS demonstrated good anti-biofilm activity against various bacteria. This study provides a new approach to the prevention and control of pathogenic bacterial biofilm.

## 1. Introduction

*Yersinia enterocolitica* (*Y. enterocolitica*) is a Gram-negative bacterium distributed worldwide and is found in soil, water, animals, and various food products. Pigs are generally considered to be the main hosts of this bacterium [1,2]. *Y. enterocolitica* can form a biofilm on the surface of food, posing a food safety risk [3]. Infections caused by *Y. enterocolitica* range from self-limited enteritis to life-threatening systemic infections [4]. *Y. enterocolitica* strains that are able to form biofilms have been isolated from infected children [3]. *Y. enterocolitica* can adapt and survive under various environmental stresses [5,6]. Therefore, in response to changes in the environment, *Y. enterocolitica* changes itself to adapt to the environment, which promotes its survival and growth within host tissues [7,8].

Outer membrane vesicles (OMVs) are spherical nanostructures produced during the growth of Gram-negative bacteria [9]. The presence of OMVs has also been reported in *Yersinia pestis* [10]. When bacteria respond to changes in the external environment, they produce corresponding stress responses. The formation of OMVs is affected by various conditions, such as antibiotics, temperature, H_2_O_2_, sulfur, UV, and pressure [11,12,13,14]. However, the formation of *P. aeruginosa* OMVs is not affected by increases in temperature [13].

OMVs mainly contain proteins (outer membrane proteins and periplasmic proteins) [15,16], nucleic acids (RNA and DNA) [17,18], lipopolysaccharide (LPS) [19,20], etc. Many biological functions that can enhance the survival chances of bacteria have been associated with OMVs, including defense against phages [21], horizontal gene transfer [22,23,24], host immune regulation [25], and biofilm formation [26,27]. A previous study demonstrated the presence of OMVs in *H. pylori* biofilm [26]. Another study reported that the protein AlpB in OMVs was secreted by *H. pylori* and affected biofilm formation [27]. In addition, the study found that the PAAP (*P. aeruginosa* aminopeptidase) in OMVs was more effective than the free substances [28]. The effects of OMVs secreted by different bacteria on biofilm formation were significantly different. The OMVs derived from *P. putida* [29], *V. cholerae* [30], *B. multivorans* [31], *F. tularensis,* and *Aeromonas* [32] significantly increased biofilm formation, whereas the OMVs derived from *P. aeruginosa*, *B. thailandensis* and Myxobacteria [33] promoted the dispersion of biofilm, resulting in weakened biofilm formation. However, no association between *Y. enterocolitica* OMVs and biofilm formation has been explored. This study focused on the association between *Y. enterocolitica* OMVs and biofilm.

With regard to the relationship between OMV biogenesis and temperature, we investigated whether OMV production was affected by temperature. Our data showed that temperature could affect the yield, protein profile, and content of the LPS in *Y. enterocolitica* OMVs. The results of the co-culture of OMVs and bacteria demonstrated *Y. enterocolitica* OMVs inhibited biofilm formation by various bacteria. Finally, we analyzed the inhibition of *Y. enterocolitica* biofilm formation by OMVs treatment.

## 2. Materials and Methods

### 2.1. Bacterial Strains and Culture Condition

The *Y. enterocolitica* strains used in this study are listed in Table 1. These strains were grown in LB broth and placed in an incubator at 28 °C. Then, samples of *Y. enterocolitica* Y1803 were cultured in LB broth for 24 h at 15 °C, 28 °C, and 37 °C. *Y. enterocolitica* was grown to a stationary phase (OD_600_ 1.0), and other bacteria were grown in LB broth at 37 °C for 24 h.

### 2.2. Quantification of Biofilm Formation

Quantification of biofilm was performed in microtiter plates as described previously [34] with slight modifications. Biofilm formation by *Y. enterocolitica* was assayed using a 96-well flat-bottom polystyrene microtiter plate. *Y. enterocolitica* was grown in LB broth overnight and diluted to OD_600_ 0.05. The culture volume was 200 µL. *Y. enterocolitica* Y1083, 3445A3-1, and C2143 were incubated at 28 °C for 48 h. *Y. enterocolitica* Y1083 was incubated at 15 °C, 28 °C, or 37 °C for 48 h. The process for measuring biofilm formation was as follows. First, sterile water was used to wash the biofilm (three times), which was then fixed with methanol (15 min). Next, 0.5% crystal violet was used to treat the biofilm (15 min). Each well was washed with sterile water. Finally, 33% acetic acid was added (10 min) and measured at OD_590_.

### 2.3. Transmission Electron Microscopy

To determine whether *Y. enterocolitica* Y1083 can secret OMVs, transmission electron microscopy (TEM) was used to observe strain Y1083. Formvar- and carbon-coated TEM grids were pre-incubated with poly-L-lysine. TEM grids were placed on top of a 10 µL droplet of cells or OMVs (1 µg/µL) (10 min) and washed with PBS (2 × 10 min). Samples were subsequently fixed in 1% glutaraldehyde PBS (5 min), followed by ddH_2_O washes (7 × 1 min). Cells or OMVs were stained with 2% uranyl oxalate pH 7.0 (5 min) and methylcellulose uranyl acetate (10 min). Samples were viewed using a Hitachi H-7500 transmission electron microscope equipped with Gatan Multiscan 791 camera.

### 2.4. Scanning Electron Microscopy

To determine whether *Y. enterocolitica* Y1083 can secret OMVs, scanning electron microscopy (SEM) was used to observe strain Y1083. The cells were washed with PBS three times to remove the floating cells. The cells were fixed with glutaraldehyde overnight. Different concentrations of ethanol (30%, 50%, 70%, 90%, and anhydrous ethanol) were used for dehydration fixation. It was then replaced with tertbutyl alcohol and dried with a freeze-dryer. Finally, biofilm was observed by SEM.

SEM was used to visualize the effect of OMVs and LPS on *Y. enterocolitica* biofilm. OMVs and LPS were incubated with *Y. enterocolitica* at 28 °C for 48 h. The process was as described above.

### 2.5. Isolation and Quantification of OMVs

*Y. enterocolitica* Y1083, 3445A3-1, and C2143 were grown to a stationary phase at 28 °C. The extraction process of OMVs has been slightly modified as described by Jana Klimentová [35]. Briefly, the same volume of bacterial culture supernatant (1200 mL) was collected, and the cells were removed by centrifugation (8000× *g*, 30 min, 4 °C). The supernatants were filtered (0.22 µm pore size) to remove residual cells. Then, the cell-free supernatants were concentrated using an ultrafiltration tube (100 kDa molecular weight cut-off, Millipore). Subsequently, the supernatants were subjected to ultracentrifugation (150,000× *g* for 3 h at 4 °C) to produce pellet OMVs (Beckman Coulter Optima 75XP ultracentrifuge with an SW 100Ti rotor). The OMV pellets were resuspended in 0.01 M phosphate-buffered saline (PBS). Finally, the OMV pellets were filter sterilized (0.22 µm pore size) and stored at −80 °C. The OMVs of strain Y1083 secreted at 15 °C, 28 °C, and 37 °C were expressed as 15 °C-OMVs, 28 °C-OMVs, and 37 °C-OMVs, respectively.

The protein concentration of OMVs was determined by using bicinchoninic acid (BCA) assay (Bradford assay kit, Beyotime, Shanghai, China). The OD_562_ was measured by using a Hitachi U-2010 spectrophotometer (Hitachi High-Tech, Tokyo, Japan). In experiments, all OMVs were normalized to a protein concentration of 600 µg/mL.

### 2.6. SDS-PAGE and Western Blot

The OMVs were separated using SDS-12% polyacrylamide gel electrophoresis (SDS-PAGE). Proteins were visualized using the silver staining method as described previously but with minor modifications [36,37]. Firstly, the protein gel was fixed (20 mL ethanol, 25 mL ultrapure water, 5 mL glacial acetic acid) and sensitized (15 mL ethanol, 35 mL ultrapure water, 0.1 g sodium thiosulfate, 3.4 g sodium acetate) for 30 min. The protein gel was washed with sterile water (3 × 5 min) and then treated for 20 min with silver nitrate staining solution (0.0625 g silver nitrate powder, 25 mL ultrapure water, 10 µL formaldehyde). The protein gel was again washed with sterile water (2 × 30 s). Color development solution (1.25 g sodium carbonate, 50 mL ultrapure water, 10 µL formaldehyde) was used for 5–10 min for color development. Finally, a stop solution (0.73 g ethylenediaminetetraacetic acid, 50 mL ultrapure water) was added for 10 min to stop the color development. The protein gel was observed and photographed. In addition, the OMVs were separated using SDS-PAGE. Nonspecific binding was blocked by incubation of the membranes in 5% non-fat milk powder for 1 h at room temperature (RT). Membranes were incubated with primary antibody (Rabbit anti-OmpF antibodies, polyclonal, 1:2000, Abcam) for 1 h at RT, followed by washing in PBS-T (0.01 M PBS containing 0.05% Tween-20) [38]. The membranes were treated with Horseradish Peroxidase-conjugated anti-rabbit IgG (1:10,000) for 1 h at RT. Finally, the membrane was incubated with ECL, and chemiluminescence was detected by using the BIO-RAD ChemiDox XRS Imaging systems (Bio-Rad, CA, USA).

### 2.7. Effect of OMVs on Biofilm Formation

To investigate the effect of OMVs on biofilm formation, *Y. enterocolitica* was co-incubated with 0, 3, 6, 12, and 24 μg OMVs at 28 °C for 48 h. Firstly, we normalized the concentration of OMVs to 600 μg/mL. The mixtures were incubated for 48 h at 28 °C. The biofilm was measured as described in Section 2.2. Similarly, OMVs were co-cultured with *S. aureus* and *S. enteritidis*, as described in Section 2.2. In addition, *S. enteritidis* strains were cultured at 37 °C for 72 h, and *S. aureus* strains were cultured at 37 °C for 24 h.

### 2.8. Growth Assay

Different concentrations of OMVs and *Y. enterocolitica* Y1083, 3445A3-1, and C2143 strains were mixed in a 96-well plate and cultured at 28 °C for 48 h. The bacterial concentration was measured at OD_600_. The blank values were removed, and the growth curve of *Y. enterocolitica* was plotted.

### 2.9. Enzymatic Hydrolysis

The OMVs were lysed using ultrasonic waves [39]. Then, intact and lysed OMVs were co-cultured with *Y. enterocolitica* 3445A3-1. To remove proteins, the DNA and RNA of intact and lysed OMVs were treated with proteinase K (50 µg/mL), RNase A (100 µg/mL), or DNase I (100 µg/mL) at 37 °C. Finally, OMVs processed as described above were co-cultured with *Y. enterocolitica* 3445A3-1. In addition, as a blank control, PBS was treated in the same way.

### 2.10. LPS Extraction from OMVs

The LPS in OMVs was extracted by a two-step extraction method, which eliminated contamination with proteins and nuclear acids. LPS was extracted by hot phenol-water method as described previously with minor modifications [40]. OMVs were resuspended with distilled water. The same volume of 90% phenol and OMVs were mixed and incubated at 68 °C for 1 h. After centrifugation at 3000× *g*, 4 °C for 30 min, the upper water phase was collected. The phenol in the upper water phase was removed by dialysis, and the crude LPS was concentrated with PEG20000. Multiple enzymes were used to remove proteins and nucleic acids in crude LPS [41]. First, DNase I (100 µg/mL) and RNase A (50 µg/mL) were added to the crude LPS and incubated at 37 °C for 4 h. Then, protease K (100 µg/mL) was added to the LPS and incubated at 55 °C for 2 h. A boiling water bath was used for 10 min. The supernatant collected after centrifugation (8000× *g*, 4 °C for 30 min) was mixed with ethanol and stood overnight at −20 °C. After centrifugation, the precipitate, LPS, was collected. LPS concentration was measured by phenol sulfuric acid method. LPS bands were visualized by silver staining method as described by Michael Frahm [42].

### 2.11. Motility Assay

Swarming motility experiments were performed in LB plates containing 0.2% agar, as described previously [43]. *Y. enterocolitica* 3445A3-1 was grown overnight in LB broth at 28 °C, diluted to OD_600_ of 0.1. PBS and OMVs/LPS were mixed with strain 3445A3-1 for the same volume, respectively. Into swarming agar plates, 1 μL mixed sample was inoculated and incubated for 48 h at 28 °C. The colony diameter was measured and compared with the diameter of the control colony to determine the percent of reduction of the area of swarming motility.

### 2.12. Expression Analyses of Biofilm-Related Genes by RT-qPCR

The total RNA was extracted from the biofilm in LB medium using the Bacterial RNA Extraction Mini Kit (Mabio, Guangzhou, China). The PrimeScript™ RT reagent Kit with gDNA Eraser (Perfect Real Time) (Takara, Beijing, China) was used in this experiment to reverse-transcribe RNA into cDNA in the presence of random primers, and cDNA amplification was completed in one step in the same reaction system. RT-qPCR was performed in a Light Cycler 96 (Roche, Basel, Switzerland) using SYBR Green and the specific primers listed in Table 2. Reactions were performed in triplicate, and the 16S rRNA gene was used as a reference for normalization.

### 2.13. Effect of LPS on Biofilm Formation

To investigate the effect of LPS isolated from OMVs on biofilm formation, LPS was co-cultured with *Y. enterocolitica* 3445A3-1. Firstly, the bacterial liquid concentration was adjusted to OD_600_ 0.05. Then, 0, 0.05, 0.1, 0.2, and 0.4 µg quantities of LPS were co-cultured with *Y. enterocolitica* 3445A3-1. The culture volume was 200 µL. Finally, strain 3445A3-1 was cultured in a 96-well flat-bottom polystyrene microtiter plate at 28 °C for 48 h.

Bacteria (Table 1) were cultured in LB broth at 37 °C. A quantity of 0.2 µg LPS was co-cultured with *S. enteritidis*, *E. coli*, *S. aureus,* and *L. monocytogenes*. Then, the bacterial liquid concentration was adjusted to OD_600_ 0.05. A 96-well flat-bottom polystyrene microtiter plate was used to culture the bacteria at 37 °C for 24 h. The biofilm was measured using the crystal violet method. Biofilm grown without LPS was used as a control.

### 2.14. Statistical Analysis

Statistical analysis used GraphPad Prism version 8.0.1. For multiple comparisons, one- or two-way analysis of variance (ANOVA) was used, followed by Tukey’s multiple comparisons test (ns, *p* > 0.05, *, *p* < 0.05, **, *p* < 0.01, ***, *p* < 0.001, ****, *p* < 0.0001) in figures, respectively.

## 3. Results

### 3.1. Characterization of Y. enterocolitica OMVs

Firstly, the biofilms of *Y. enterocolitica* Y1083, 3445A3-1, and C2143 were measured. *Y. enterocolitica* Y1083, 3445A3-1, and C2143 were grown at 28 °C for 48 h. The results showed that strain Y1083 had the strongest biofilm formation capability, followed by strain 3445A3-1 and strain C2143 (Appendix A). *Y. enterocolitica* Y1083 was incubated at 15 °C, 28 °C, or 37 °C for 48 h. The results showed that the biofilm formation capability of strain Y1083 was strongest at 15 °C (Appendix A). To confirm whether OMVs can be secreted by *Y. enterocolitica* Y1083, SEM and TEM were used to analyze cells at stationary phases. The results demonstrated that OMVs were visible on the surface of the cells. (Appendix A). The vesicles secreted by the bacteria had a large size distribution. Appendix A shows OMVs secreted from two cells, one of which is larger, whereas the other is smaller and similar to the size of the vesicles shown in Appendix A. Therefore, it can be seen that *Y. enterocolitica* Y1083 may also secrete some extremely large vesicles.

It has been reported that the shape and yield of OMVs derived from *F. tularensis* were subject to change at different temperatures [44]. To investigate the effect of temperature on the secretion of OMVs by *Y. enterocolitica*, OMVs were isolated from strain Y1083 under different temperatures. At 15 °C, 28 °C, and 37 °C, the biofilm-forming ability of *Y. enterocolitica* Y1083 gradually decreased (Appendix A). *Y. enterocolitica* Y1083 OMVs secreted at 15 °C, 28 °C, and 37 °C were expressed as 15 °C-OMVs, 28 °C-OMVs, and 37 °C-OMVs, respectively. Then, 15 °C-OMVs, 28 °C-OMVs, and 37 °C-OMVs were characterized. Figure 1A shows that the morphology of 15 °C-OMVs, 28 °C-OMVs, and 37 °C-OMVs was similar and spherical. OmpF (outer membrane porin F) was abundant in the outer membrane [45,46]; therefore, anti-OmpF antibodies were used to detect the presence of OMVs [47]. There was a correct protein band (42 kDa) in these OMVs (Figure 1B). The results demonstrated the presence of OMVs. BCA assay was used to quantify the protein concentration of the OMVs [48]. At 15 °C and 28 °C, the yield of protein was higher (1262 ± 143.6 µg/mL and 1054 ± 92.06 µg/mL, respectively), and at 37 °C, the yield of protein decreased (1054 ± 92.06 µg/mL) (Figure 1C).

A previous study reported that the components of *Campylobacter jejuni* OMVs were affected by temperature [49]. We analyzed OMVs using SDS-PAGE. As shown in Figure 1D, the protein profile of 15 °C-OMVs was very similar to the bands of 28 °C-OMVs and 37 °C-OMVs. However, a unique protein band (about 45–60 kDa) was found in 15 °C-OMVs. A unique band of about 50 kDa can be found in three types of OMVs. Compared with the other conditions, this protein was most abundant in the 28 °C-OMVs, with a clearly visible band, whereas for the 15 °C-OMVs and 37 °C-OMVs, the bands were very blurred. In summary, temperature can affect the yield and protein composition of *Y. enterocolitica* OMVs, but it does not affect their morphology.

### 3.2. Effects of Y. enterocolitica OMVs on Biofilm Formation

It was reported that *H. pylori* OMVs could enhance the formation of *H. pylori* biofilm [26]. To verify the effect of *Y. enterocolitica* OMVs on biofilm formation, 15 °C-OMVs, 28 °C-OMVs, and 37 °C-OMVs were individually co-cultured with *Y. enterocolitica* 3445A3-1. The biofilm formation of strain 3445A3-1 was weaker than that of Y1083. As shown in Figure 2, the concentrations of 15 °C-OMVs, 28 °C-OMVs, and 37 °C-OMVs were 15 µg/mL, 30 µg/mL, and 60 µg/mL, respectively, and the biofilm formation of *Y. enterocolitica* 3445A3-1 was inhibited. Moreover, this inhibition effect was not dosage-dependent. Therefore, we speculate that there are differences in concentrations of the substances in OMVs that inhibit biofilm formation. Next, the mechanism by which 28 °C-OMVs (referred to as OMVs) inhibit biofilm formation will be explored.

To further investigate the relationship between OMVs and *Y. enterocolitica*, OMVs were co-cultured with *Y. enterocolitica* Y1083, 3445A3-1, and C2143 strains. We have treated bacteria with OMVs at 60 or 120 μg/mL. The addition of OMVs did not affect the growth of the Y1083, 3445A3-1, and C2143 strains (Figure 3A). However, the biofilm formation of the Y1083, 3445A3-1, and C2143 strains decreased following treatment with OMVs (Figure 3B). The results proved that the inhibitory effect of *Y. enterocolitica* on biofilm formation is not caused by the inhibition of bacterial growth. Similarly, OMVs derived from the 3445A3-1 and C2143 strains also decreased *Y. enterocolitica* biofilm formation (Appendix A). These results demonstrated that the inhibitory effect of *Y. enterocolitica* OMVs on biofilm may be common in *Y. enterocolitica*.

To evaluate the role of OMVs in the biofilm formation stage, OMVs were co-cultured with the 3445A3-1 strain for 3, 6, 12, 24, 48, 72, and 96 h. We found that OMVs decreased the biofilm formation at the initial stage of the biofilm and the inhibitory effect lasted 96 h (Appendix A). To investigate whether OMVs decrease biofilm formation by degrading the biofilm, OMVs were added when the 3445A3-1 strain had been grown for 0, 12, and 24 h. The results showed that OMVs could inhibit biofilm formation when added to the 3445A3-1 strain at 0 h of growth. However, the addition of OMVs after the 3445A3-1 strain had been growing for 12 and 24 h did not clear the *Y. enterocolitica* biofilm that had formed (Appendix A). When added early, *Y. enterocolitica* OMVs can significantly inhibit biofilm formation and have a long-term inhibitory effect.

Naturally occurring biofilms are composed of multiple bacterial species, and the interaction between species can affect the development and structure of the community. *S. enteritidis* and *S. aureus* were selected as representative Gram-negative and Gram-positive bacteria. To investigate the effect of *Y. enterocolitica* OMVs on the formation of *S. enteritidis* and *S. aureus* biofilms, OMVs were co-cultured with *S. enteritidis* and with *S. aureus*. The results showed that biofilm formation by *S. enteritidis* 51-71 and 48-1 strains was significantly inhibited, whereas biofilm formation by the L38 strain was decreased by OMVs, but the inhibitory effect was not statistically significant (Figure 3C). Biofilm formation by *S. aureus* strains 2743-1, 4074, and 1048 was inhibited (Figure 3D). In conclusion, OMVs inhibited biofilm formation at the initial stage of biofilm formation, and the effect persisted for several days but had no effect on bacterial growth. In addition, *Y. enterocolitica* OMVs inhibited biofilm formation by both *Y. enterocolitica* strains and other bacteria (*S. enteritidis* and *S. aureus*).

### 3.3. Inhibition of Biofilm Formation by LPS

It has been reported that OMVs protect their own components to a certain degree [50]. Therefore, we speculated that the integrity of OMVs might be relevant to their functions. OMVs were, therefore, treated with ultrasound to destroy their integrity (Figure 4A). TEM showed that fragments of fractured vesicles were present (Appendix A). Both the intact and broken OMVs inhibited biofilm formation (Figure 4B). Figure 4B showed that the loss of OMV integrity did not affect its inhibitory effect. To evaluate whether the proteins and nucleic acids obtained from OMVs play a role in the process of biofilm formation. Proteinase K, DNase I, or RNase A were used to clear proteins and nucleic acids on the surface of OMVs, respectively (Figure 4B). The results showed that OMVs that were intact following enzyme treatment could significantly decrease biofilm formation (Figure 4B). To eliminate the proteins, DNA or RNA localized inside OMVs, proteinase K, DNase I, and RNase A were each incubated with broken OMVs. After enzyme treatment, broken OMVs also decreased biofilm formation. Compared with intact OMVs, the inhibitory effect of broken OMVs was not significantly different. In addition, biofilm was significantly inhibited after the addition of protease K or DNase I (Figure 4B). These results indicate that proteins and eDNA may constitute the structure of strain 3445A3-1 biofilm. The addition of RNase A did not reduce biofilm formation. In summary, the biofilm formation of *Y. enterocolitica* was not affected by the OMV proteins, DNA or RNA.

A previous study reported that OMVs contain LPS components [51]. Therefore, we investigated whether LPSs are involved in biofilm formation. Of the three OMVs with the same protein concentration, the LPS content in the 15 °C-OMVs (36.36 ± 3.67 µg/mL) was the highest, followed by the 28 °C-OMVs (25.97 ± 2.75 µg/mL), and the lowest content was found in the 37 °C-OMVs (16.72 ± 4.08 µg/mL) (Figure 4C). LPSs derived from 15 °C-OMVs, 28 °C-OMVs, and 37 °C-OMVs were observed after silver staining (Appendix A). SDS-PAGE revealed that the LPS was R-type because of the presence of bands instead of smears [52]. In order to investigate the effect of LPS on biofilm formation, strain 3445A3-1 was treated with LPS. The results showed that LPS derived from three kinds of OMVs inhibited biofilm formation at a concentration of 0.1 µg (Figure 4D).

To evaluate the sustained inhibitory effect of LPS on biofilms, 0.2 μg LPS was co-cultured with *Y. enterocolitica* 3445A3-1 for periods of 3, 6, 12, 24, 48, 72, and 96 h. We extracted amounts of LPS ranging from 2000 µg to about 30 µg. In the previous experiments, 12 µg OMVs were generally used, so the amount of LPS added in the later experiments was selected to be 0.2 µg. The results showed that compared with the control, LPS significantly inhibited biofilm formation from 0 to 96 h. In order to determine whether LPS degrades biofilm, samples of biofilm grown for 0, 6, 12, and 24 h were each treated with LPS. The results showed that LPS only had the effect of inhibiting biofilm formation in the early stage of biofilm formation and had no clearing effect on the biofilm that was already formed (Figure 4E,F).

The morphology of *Y. enterocolitica* 3445A3-1 biofilm observed using SEM is consistent with the biomass measured using crystal violet staining (Figure 5). The untreated biofilm (control group) was dense with strong intercellular adhesion. Biofilm treated with OMVs showed less cell aggregation and intercellular adhesion. Biofilm treated with LPS revealed a single-cell distribution and almost no aggregation. Taken together, the inhibitory effect of *Y. enterocolitica* OMVs on biofilm formation was attributed to the LPS it contains.

### 3.4. LPS Inhibited the Expression of Biofilm-Related Genes

Biofilm formation is a complex process that involves many factors, such as bacterial motility [53]. Swarming motility is considered a key factor in enhancing bacterial colonization of surfaces, so we investigated the effect of OMVs and LPS on the motility of *Y. enterocolitica* 3445A3-1. Compared with the control (without treatment), the addition of OMVs and LPS reduced the bacterial range of motion from 4.90 ± 1.15 cm to 1.77 ± 0.25 cm and 1.73 ± 0.31 cm, respectively (Figure 6A,B). The results demonstrated that OMVs and LPS inhibited the motility of *Y. enterocolitica*.

We suspected that the motility impairment effect of LPS on *Y. enterocolitica* might be linked to the flagella function as this motoric protein plays a substantial role in the motility of *Y. enterocolitica* and its biofilm. It has been reported that the gene *pgaABCD* can regulate biofilm formation [54,55]. Therefore, we also investigated the effect of LPS on the expression of genes *pgaABCD* in biofilm. Our RT-qPCR results proved that the mRNA level of the motility- and biofilm-related genes *pgaA*, *pgaB*, *pgaC*, *pgaD*, *motB*, and *flhBD* was significantly inhibited compared with the control. The expression levels of *pgaA*, *pgaB*, *pgaC*, *motB*, *flhB*, and *flhD* genes were 59.1%, 61.9%, 68.4%, 65.8%, 69%, and 70.9%, respectively, of the level observed in control cells (*Y. enterocolitica* biofilm cells without LPS treatment). These results suggest that the inhibition of *Y. enterocolitica* biofilm formation may be caused by a reduction in cell motility and the expression of genes related to biofilm formation.

### 3.5. LPS Inhibit Biofilm Formation with Broad-Spectrum Ability

To evaluate the spectrum of the inhibitory effect of LPS on bacterial biofilms, LPS was used in biofilm assays with the Gram-negative (*E. coli* and *S. enteritidis*) and Gram-positive (*L. monocytogenes* and *S. aureus*) bacteria. Figure 7A shows that biofilm formation by *E. coli* (3724-3, 1553-5, and 230-1), *S. enteritidis* (51-71), *L. monocytogenes* (1516-2, 2919-1, and 948-1), and *S. aureus* (4074 and 1048) was inhibited by LPS isolated from *Y. enterocolitica* OMVs. The inhibition rate of LPS on *E. coli* (1553-5 and 230-1) and *L. monocytogenes* (1516-2, 2919-1, and 948-1) biofilm formation reached about 90% (Figure 7B). However, the inhibitory effect on *E. coli* (3724-3), *S. enteritidis* (L38 and 51-71), and *S. aureus* (4074 and 1048) biofilms were not strong. In addition, the growth of *Y. enterocolitica*, *E. coli*, *L. monocytogenes*, and *S. enteritidis* was slightly promoted by LPS (Appendix A). This indicates that LPS has a long-lasting and broad-spectrum anti-biofilm effect in multiple bacterial species.

## 4. Discussion

In this study, the formation of *Y. enterocolitica* OMVs and biofilm was analyzed at different temperatures. Previous studies found that the formation of OMVs is regulated by temperature [56]. *P. aeruginosa* formed the most robust biofilm at 20 °C; however, as the temperature increased to 25 °C, biofilm formation rapidly decreased [57].

At 15 °C, *Y. enterocolitica* strain Y1083 had the highest secretion of OMVs, as well as the highest biofilm formation. Previous studies had revealed the presence of OMVs in the *H. pylori* biofilm matrix, and it was speculated that OMVs could promote the formation of biofilm [26]. As we know, proteins, extracellular DNA (eDNA), and polysaccharides are among the scaffold components of biofilm [58,59,60]. OMVs have been shown to contain proteins, DNA, and LPS [16,17]. Based on these previous findings, we speculate that the OMVs secreted by *Y. enterocolitica* may affect the formation of *Y. enterocolitica* biofilm.

In our study, we found that 15 °C-OMVs, 28 °C-OMVs, and 37 °C-OMVs significantly inhibited biofilm formation. Therefore, we speculated that the components in *Y. enterocolitica* OMVs were not affected by temperature. In addition, the strains that we isolated from food were obtained after enrichment at 28 °C. To study the role of bacterial OMVs in biofilm formation in the environment more accurately, 28 °C-OMVs were used for subsequent experiments.

*Y. enterocolitica* OMVs inhibited biofilm formation in itself and in other *Y. enterocolitica* strains. However, OMVs treatment did not affect bacterial growth. This implies that the inhibition of biofilm formation was not due to the bactericidal or bacteriostatic effects of OMVs. We also found that *Y. enterocolitica* OMVs could decrease biofilm formation by *S. enteritidis* and *S. aureus*. This phenomenon indicates that OMV components can be involved in biofilm formation by various bacteria.

We further investigated why OMVs inhibited biofilm formation. Studies have shown that OMVs secreted by *P. aeruginosa*, which contain PAAP, cause protease-mediated biofilm detachment, leading to changes in matrix and colony composition [28]. We predicted that a certain protein localized on the OMVs might be involved in the formation of biofilm. We found the proteins packaged in *Y. enterocolitica* OMVs did not affect biofilm formation. In addition, we found several proteins (about 45–60 kDa) that were only present in the 15 °C-OMVs. However, LC-MS results showed that the specific proteins (about 45–60 kDa) in the 15 °C-OMVs were not associated with biofilm formation. Studies have shown that eDNA is mainly associated with the membrane surfaces of OMVs [61]. It has also been reported that sRNA can regulate biofilm formation [62]. However, we demonstrated that DNA and RNA did not inhibit biofilm formation by *Y. enterocolitica*.

LPS is an essential structural molecule in the outermost part of the cell envelope, and it consists of three distinct domains: core oligosaccharides, lipid part A, and O-antigens [63,64]. An association between *Y. enterocolitica* LPS and biofilm formation has been reported [65]. It was shown that the deletion of the *Y. enterocolitica waaF* gene truncated the structure of LPS and generated a deep, rough LPS. The truncated LPS increased the cell surface hydrophobicity and repressed cell motility and biofilm formation. In our study, we found that LPS isolated from OMVs inhibited *Y. enterocolitica* motility. LPS contains polysaccharides, which have been reported to inhibit the formation of bacterial biofilms [66]. The group II envelope polysaccharide (a soluble polysaccharide) is secreted by *E. coli* and prevents biofilm formation by a broad spectrum of Gram-positive and Gram-negative bacteria. The group II envelope polysaccharide induces physical and chemical changes on the abiotic surface, thereby inhibiting bacterial colonization on the surface and reducing biofilm formation [67].

Bacterial motility and the expression of the biofilm-related genes *pgaABC*, *motB,* and *flhBD* were inhibited by LPS (Figure 6A,C). The pgaABCD locus is an important factor in biofilm formation by *E. coli* because it mediates cell-to-cell and cell-to-surface adhesion in biofilm [68]. Proteins FlhB and FlhC are both proteins synthesized by flagella. Bacterial motility is associated with biofilm formation [69,70]. Bacterial flagella can regulate the movement of bacteria and thus affect the formation of biofilm. The main manifestations are as follows: flagella promote the initial adhesion of bacteria to the surface, and during biofilm development, locomotion promotes the spread of bacteria along the surfaces. Based on the results of this study, it is speculated that LPS may affect bacterial motility by inhibiting the flagella of *Y. enterocolitica* and leading to the influence of the adhesion and development of *Y. enterocolitica* in the initial stage of biofilm formation. In our study, the exogenous addition of self-secreted OMVs and LPS also significantly inhibited biofilm formation. We hypothesized that OMVs and LPS adhered to the surface of the object during the initial addition, resulting in the inability of the bacteria to adhere to the object surface, resulting in a reduction in the formation of bacterial biofilm.

The addition of LPS isolated from *V. vulnificus* inhibited the formation of *V. vulnificus* biofilm [71]. In addition, *V. vulnificus* LPS without lipid A-core oligosaccharide also inhibited the biofilm formation of multiple Gram-negative bacteria. However, an LPS fraction extracted from a mutant unable to produce O-antigen polysaccharides did not have an inhibitory effect. Studies have demonstrated that O-antigens are necessary for many bacteria to swim and swarm [72,73]. The loss of O-antigens in *P. aeruginosa* will lead to a decrease in its motor ability [74]. In this study, we demonstrated that LPS reduced cell motility. Based on this, we hypothesized that the O-antigen of LPS may play a role in decreasing biofilm formation.

This study elucidates the relationship between *Y. enterocolitica* OMVs and biofilm formation and may provide useful information for the prevention of *Y. enterocolitica* biofilm infection.

## 5. Conclusions

*Y. enterocolitica* is a common food-borne pathogen that can pollute foods. There were differences in the yield, protein profile, and content of LPS of *Y. enterocolitica* OMVs produced at 15 °C, 28 °C, and 37 °C. It was found that *Y. enterocolitica* OMVs inhibited biofilm formation but did not affect bacterial growth. In addition, *Y. enterocolitica* OMVs were found to have antibiofilm activity against *S. enteritidis* and *S. aureus*. Biofilm formation by *Y. enterocolitica* was inhibited by LPS in OMVs, rather than by protein, DNA, or RNA. We demonstrated that LPS inhibited the motility of *Y. enterocolitica* and the expression of biofilm-related genes (*pgaABC*, *motB*, *flhBD*). Finally, our results showed that LPS has a long-lasting and broad-spectrum antibiofilm effect in multiple bacterial species.

## Figures and Tables

**Figure 1 microorganisms-10-02357-f001:**
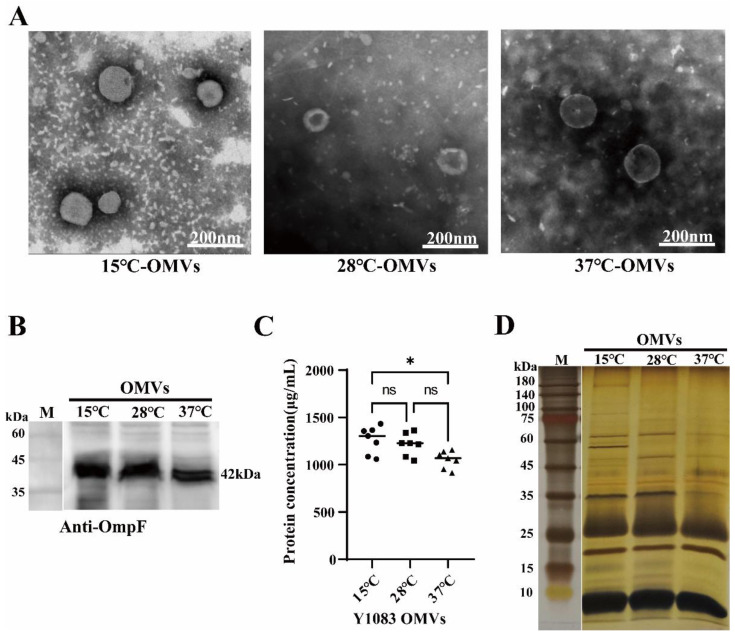
The formation of *Y. enterocolitica* Y1083 OMVs at different temperatures. (**A**) TEM was used to observe the morphology of OMVs. (**B**) Western blot analysis OmpF (42 kDa) in OMVs isolated from strain Y1083 which was cultured at 15 °C, 28 °C, and 37 °C. SDS-PAGE was probed with anti-OmpF antibodies. (**C**) The OMV production concentrations were measured using a BCA kit. One-way ANOVA was used, followed by Tukey’s multiple-comparison test using GraphPad Prism version 8.0.1 to assess significance. ●, ◼ and ▲ represent culture at 15 °C, 25 °C and 37 °C, respectively. Error bars indicate the standard deviations of seven measurements. ns, *p* > 0.05, *, *p* < 0.05. (**D**) OMV proteins were observed using SDS-PAGE. The protein gel was treated with silver staining.

**Figure 2 microorganisms-10-02357-f002:**
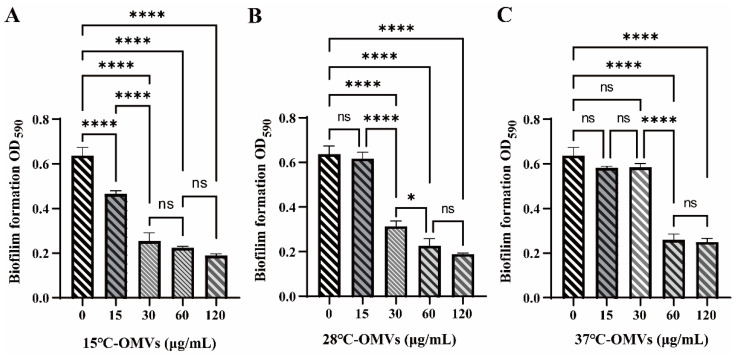
Effects of different OMVs on biofilm formation. (**A**–**C**) 15 °C-OMVs 28 °C-OMVs and 37 °C-OMVs at concentrations of 0, 15, 30, 60, and 120 μg/mL were each co-cultured with strain 3445A3-1 at 28 °C for 48 h. One-way ANOVA was used, followed by Tukey’s multiple-comparison test using GraphPad Prism version 8.0.1 to assess significance. Error bars indicate the standard deviations of seven measurements. ns, *p* > 0.05, *, *p* < 0.05, ****, *p* < 0.0001.

**Figure 3 microorganisms-10-02357-f003:**
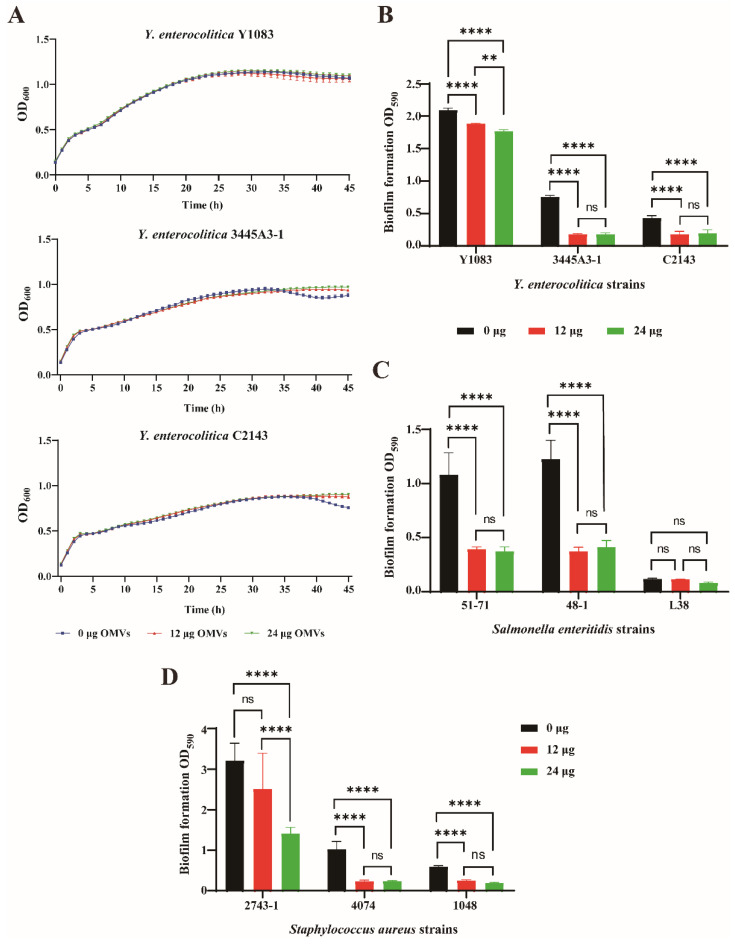
*Y. enterocolitica* OMVs inhibited the formation of bacterial biofilms. (**A**) The concentration of *Y. enterocolitica* Y1083, 3445A3-1, and C2143 grown in the 96-well plate was measured using a growth curve meter. (**B**) Biofilm formation of the Y1083, 3445A3-1, and C2143 strains was measured using crystal violet staining. (**C**) OMVs and the *S. enteritidis* 51-71, 48-1, and L38 strains were incubated at 37 °C for three days. (**D**) OMVs and the *S. aureus* 2743-1, 4074, and 1048 strains were incubated at 37 °C for one day. One-way ANOVA was used, followed by Tukey’s multiple-comparison test using GraphPad Prism version 8.0.1 to assess significance. Error bars indicate the standard deviations of seven measurements. ns, *p* > 0.05, **, *p* < 0.01, ****, *p* < 0.0001.

**Figure 4 microorganisms-10-02357-f004:**
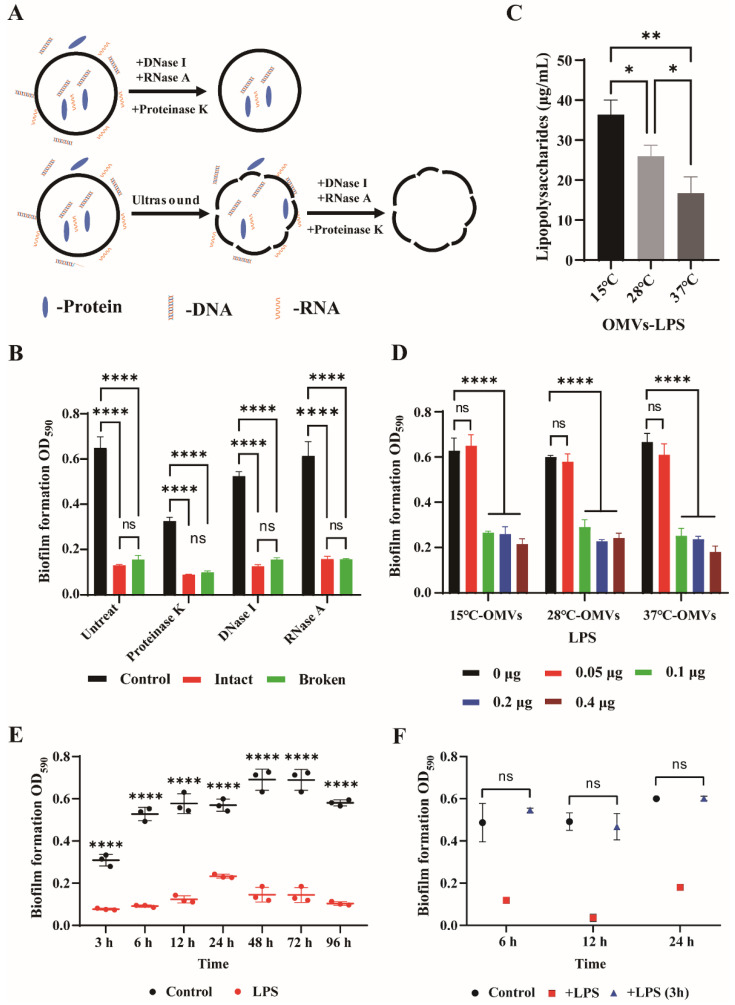
The role of various components in *Y. enterocolitica* OMVs. (**A**,**B**) OMVs were lysed using ultrasound. Intact and lysed OMVs were separately incubated with protease K (50 µg/mL), DNase I (100 µg/mL), or RNase A (100 µg/mL) at 37 °C for 1 h. After protease K (50 µg/mL), DNase I (100 µg/mL), or RNase A (100 µg/mL) treatment, PBS (0.01 M), and OMVs were co-cultured with bacteria, and biofilm formation was measured. The control was a sample treated without addition of OMVs (**C**) LPSs were measured using the phenol sulfuric acid method. (**D**) 0, 0.05, 0.1, 0.2, and 0.4 µg LPS were co-cultured with strain 3445A3-1 at 28 °C for 48 h. (**E**) After adding 0.2 μg LPS and *Y. enterocolitica* 3445A3-1 for 3, 6, 9, 12, 24, 48, 72, and 96 h, the amount of biofilm formation was measured using crystal violet staining. (**F**) *Y. enterocolitica* 3445A3-1 was cultured for 0, 12, and 24 h. A quantity of 0.2 µg LPS was added at 0 h and co-cultured for 0, 12, and 24 h (square). A quantity of 0.2 µg LPS was added and co-culture for 3 h (triangle). One-way ANOVA was used, followed by Tukey’s multiple-comparison test using GraphPad Prism version 8.0.1 to assess significance. Error bars indicate the standard deviations of seven measurements. ns, *p* > 0.05, *, *p* < 0.05; **, *p* < 0.01; ****, *p* < 0.0001.

**Figure 5 microorganisms-10-02357-f005:**
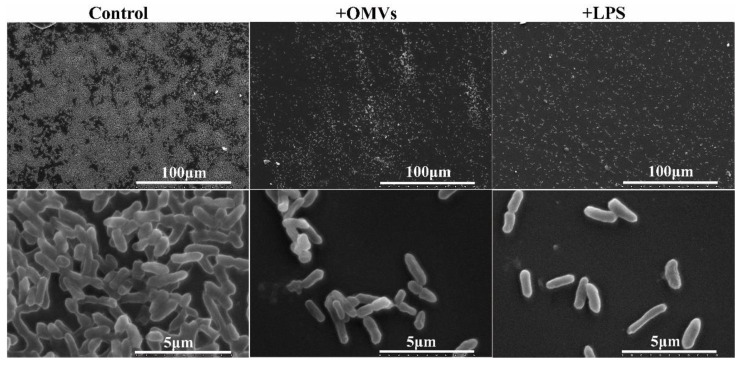
SEM was used to observe the microstructure of 3445A3-1 strain biofilm. Control means untreated biofilm; +OMVs means the biofilm of treated OMVs; +LPS means the biofilm of treated LPS.

**Figure 6 microorganisms-10-02357-f006:**
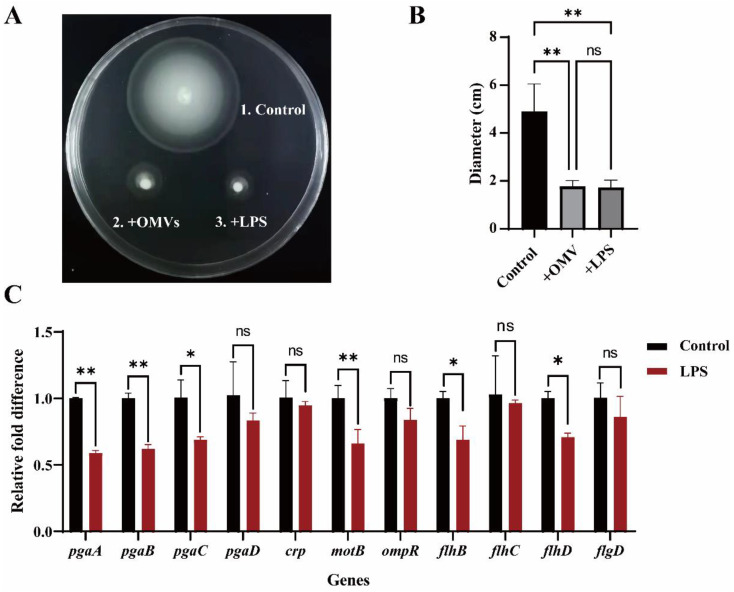
LPSs inhibit the expression of biofilm-related genes. (**A**,**B**) *Y. enterocolitica* 3445A3-1 was co-cultured with OMVs and with LPS in LB plates containing 0.2% agar for 48 h. (**C**) RT-qPCR was used to analyze the differences in the expression of biofilm-related genes. Control: Gene expression in untreated biofilms. LPS: Gene expression in biofilms after LPS treatment. One-way ANOVA was used, followed by Tukey’s multiple-comparison test using GraphPad Prism version 8.0.1 to assess significance. Error bars indicate the standard deviations of seven measurements. ns, *p* > 0.05, *, *p* < 0.05, **, *p* < 0.005.

**Figure 7 microorganisms-10-02357-f007:**
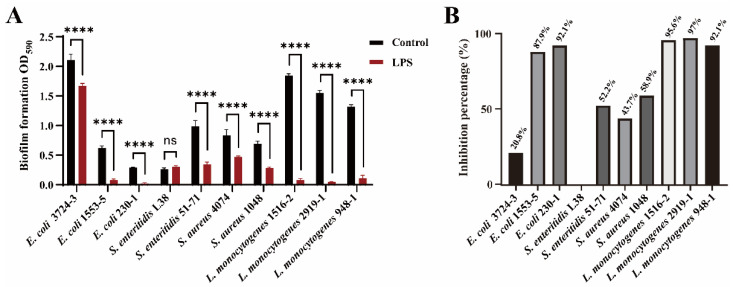
The effect of LPS on biofilm formation. (**A**) 0.2 μg LPS was mixed with *E. coli*, *S. enteritidis*, *S. aureus,* and *L. monocytogenes* in a 96-well plate and cultured at 37 °C for 24 h. (**B**) The inhibition rate of each sample biofilm formation in Figure A. Error bars indicate the standard deviations of seven measurements. ns, *p* > 0.05, ****, *p* < 0.0001.

**Table 1 microorganisms-10-02357-t001:** Experimental strains.

Bacteria	Strains	Source	Reference
*Yersinia enterocolitica*	Y1083	Beef	This study
*Yersinia enterocolitica*	3445A3-1	Mutton	This study
*Yersinia enterocolitica*	C2143	Chicken	This study
*Salmonella enteritidis*	L38	Diarrhea patient	This study
*Salmonella enteritidis*	51-71	Dumpling	This study
*Salmonella enteritidis*	48-1	Pork	This study
*Escherichia coli*	3724-3	Chicken	This study
*Escherichia coli*	1553-5	Minced meat	This study
*Escherichia coli*	230-1	Chicken	This study
*Staphylococcus aureus*	4074	Chicken	This study
*Staphylococcus aureus*	1048	Chicken	This study
*Staphylococcus aureus*	2743-1	Chicken	This study
*Listeria monocytogenes*	1516-2	Meat bun	This study
*Listeria monocytogenes*	2919-1	Enoki mushroom	This study
*Listeria monocytogenes*	948-1	Chicken	This study

**Table 2 microorganisms-10-02357-t002:** Primers used in the RT-qPCR assay.

Primer	Relevant Physiological Process	Sequences (5′→3′)	bp	Reference
*flhA*	flagellar biosynthesis protein	F: CGAAGCTGACCGAGGACTTT R: GTACCACCGCGGTTAACTCA	174	This study
*flhB*	flagellar type III secretion system protein	F: TGATTGCTCAGGGGTTGCAT R: TTGACTCGCCGCTAAACAGT	183	This study
*flhC*	flagellar transcriptional regulator	F: TGGACCTTGGTTCGCTTTGT R: ATCGGCAGATTGCGGAGAAA	174	This study
*flhD*	flagellar transcriptional regulator	F: CATGGCCGATGCATTATCGC R: ATTCCGGTGTGGATTTGCTG	159	This study
*flgD*	flagellar hook assembly protein	F: GTGACGACAACACCGTTTGG R: TGACGGTATAAGCGCCATCC	190	This study
*pgaA*	poly-beta-1,6 N-acetyl-D-glucosamine export porin	F: AGTCAGATTATCAGCGGGCG R: AATATAGGCAGAAGCGGCCC	152	This study
*pgaB*	poly-beta-1,6-N-acetyl-D-glucosamine N-deacetylase	F: ACATGGCTACCAAATGGCGA R: GTGCCGGATCTGGATCGTAG	197	This study
*pgaC*	poly-beta-1,6 N-acetyl-D-glucosamine synthase	F: CGTACCCGTTCGACCTTGAT R: ATCCGGACTCCAGTAACCGA	168	This study
*pgaD*	poly-beta-1,6-N-acetyl-D-glucosamine biosynthesis protein	F: GGGCCGATCCCGTTTAGAAT R: TCCATGCTATCGGCGAGTTC	185	This study
*motB*	flagellar motor protein	F: CGCAAAGCCAAACATGGTCA R: TTTCAACGGCGTGCGAAAAT	165	This study
*ompR*	two-component system response regulator	F: AACCTATGCCACTGACCAGC R: GGCGCAAACGTGAAATCTGT	159	This study
*crp*	cAMP-activated global transcriptional regulator	F: GCCTGTCTTCACAAATGGCG R: GACCAATTTCCTGGCGGGTA	177	This study
16S	-	F: AGCCATGCCGCGTGTGTGAAGA R: AAACCGCCTGCGTGCGATTTAC	199	This study

## Data Availability

Not applicable.

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
