# Peer review of "Yersinia enterocolitica-Derived Outer Membrane Vesicles Inhibit Initial Stage of Biofilm Formation"

_microorganisms, 2022, doi:10.3390/microorganisms10122357_

Round 1

Reviewer 1 Report

Dear Authors,

The presented manuscript concerns the biofilm formation by Y. enterocolitica strains at various temperatures. However, while reading the introduction, the reader was not properly introduced to the topic linking biofilm with the pathogenicity of this bacterium. It should be supplemented with information on the ability to form biofilm from a clinical point of view. Why is it so important. After all, this is the main reason why biofilm abilities are tested in order to fight pathogenic microorganisms.

When describing SDS-PAGE, the Authors claim that it was made in accordance with references 36 and 37, but with minor modifications. What? they should be mentioned.

The manuscript requires minor editorial corrections. The methods do not unify the centrifugation conditions. In subsection 2.10, the Authors use 3000r / min, and above, in 2.5, 8,000g. Why is it not standardized?

Should the writing of the temperature also be standardized, with or without a space?

Line 315, no "I" on DNase.

I am asking the Authors to review the work once again in terms of editing.

Best regards

Reviewer 2 Report

In this work, the authors have characterized the role of outer membrane vesicles on biofilm formation of Yersinia enterocolitica. Previous works from different organisms have shown that OMV can act both positively as well as negatively on the biofilm formation. However, this is the first time the authors have established this relationship between OMV and biofilm formation in Yersinia. Previous works have also found out that the composition of the OMVs is dependent on the temperature of growth. The authors have purified the OMVs at three different temperature and analyzed their content by running it in SDS PAGE. Indeed, at 15C the authors found a novel protein in the OMV that is missing from OMV extracted at 28 and 37C. The authors also found that OMV isolated from growth at 28C where able to inhibit the biofilm formation, but no change was observed for the bacterial growth. Moreover, the OMV were most potent when added at the beginning of the biofilm formation and may have a long-lasting inhibitory effect. The authors also found that neither the integrity of the OMVs nor the DNA, RNA or protein constituent of the OMV could affect their inhibitory function on biofilm formation. However, lipopolysaccharide, which is a constituent of the OMV could significantly alter the biofilms. Indeed, when the bacteria was co-cultured with LPS, the later could inhibit the biofilm formation of the bacteria. The authors found that LPS was able to inhibit biofilm formation by decreasing the expression of the genes involved in biofilm formation. LPS was also shown to inhibit biofilm formation of a variety of bacterial species including Escherichia coli and Salmonella.

The paper is well written and easy to follow. The materials and methods are descriptive. The work should be of interest to the readers involved in studying biofilm formation. I’ve couple of queries that I am outlining below:

1.     Did the authors characterize the novel protein that was present in the OMV at 15C but lacking from other OMVs using Mass Spec? The authors should comment on this.

2.     How does LPS inhibit the expression of these genes? The authors should explain it clearly in the discussion section.

Reviewer 4 Report

The work has a good potential, but is very badly written. English and style must be seriousely improved

I stoped the review at line 321 "Biofilm  formation of strain 3445A3-1 by LPS treatment was analyzed. "  This sentence means that "LPS treatment formed biofilm of 3445A3-1 strain" -

More minor comments are in the attachd pdf file

Major comments

1) If I undestand well, the LPS experiment was badly planned: LPS were added first to bacteria and then bacteria concentration was adjusted: that changed the concentration of LPS.  line190-193

2) I cannot understand if authors made co-cultures of Yersinia and other bacteria or pure cultures of other bacteria?

3) Agar dilution and broth dilution are the most commonly used techniques to determine the minimal inhibitory concentration (MIC) of antimicrobial agents. Authors did not applied these techniques.

4)  MIC is the lowest concentration of an antibacterial agent, which, under strictly controlled in vitro conditions, completely prevents visible growth of the test strain of an organism. In the presented work, authors have demonstrated that OMV did not affect the bacterial growth but biofilm formation.

Round 2

Reviewer 4 Report

I have a concern about some points that are highlighted in the text
